# Tunable Magnetic Hyperthermia Properties of Pristine and Mildly Reduced Graphene Oxide/Magnetite Nanocomposite Dispersions

**DOI:** 10.3390/nano10122426

**Published:** 2020-12-04

**Authors:** Erzsébet Illés, Etelka Tombácz, Zsófia Hegedűs, Tamás Szabó

**Affiliations:** Department of Physical Chemistry and Materials Science, University of Szeged, Rerrich Béla tér 1, H-6720 Szeged, Hungary; tombacz@chem.u-szeged.hu (E.T.); hegedus.zsofia95@gmail.com (Z.H.)

**Keywords:** magnetite nanoparticles, graphene oxide, graphite oxide, heteroaggregation, chemical reduction, ascorbic acid, nanocomposite dispersion, heat production, magnetic hyperthermia

## Abstract

We present a study on the magnetic hyperthermia properties of graphene oxide/magnetite (GO/MNP) nanocomposites to investigate their heat production behavior upon the modification of the oxidation degree of the carbonaceous host. Avoiding the harsh chemical conditions of the regular in situ co-precipitation-based routes, the oppositely charged MNPs and GO nanosheets were combined by the heterocoagulation process at pH ~ 5.5, which is a mild way to synthesize composite nanostructures at room temperature. Nanocomposites prepared at 1/5 and 1/10 GO/MNP mass ratios were reduced by NaBH_4_ and L-ascorbic acid (LAA) under acidic (pH ~ 3.5) and alkaline conditions (pH ~ 9.3). We demonstrate that the pH has a crucial effect on the LAA-assisted conversion of graphene oxide to reduced GO (rGO): alkaline reduction at higher GO loadings leads to doubled heat production of the composite. Spectrophotometry proved that neither the moderately acidic nor alkaline conditions promote the iron dissolution of the magnetic core. Although the treatment with NaBH_4_ also increased the hyperthermic efficiency of aqueous GO/MNP nanocomposite suspensions, it caused a drastic decline in their colloidal stability. However, considering the enhanced heat production and the slightly improved stability of the rGO/MNP samples, the reduction with LAA under alkaline condition is a more feasible way to improve the hyperthermic efficiency of magnetically modified graphene oxides.

## 1. Introduction

Magnetic nanomaterials attract rising attention in biomedical applications, such as easily controllable targeted drug-delivery vehicles for cancer therapy [1,2,3]. Superparamagnetic iron oxide, i.e., magnetite nanoparticles (MNPs) have great potential for biomedical use (MRI contrast enhancement, targeted drug delivery, etc.). Although the MNPs have inherent unique magnetic properties, they should be biocompatibilized, e.g., with carboxylate-type compounds [4] for their application in living systems. Beside carrying and releasing the active compound into the tumor tissues, magnetic particles are able to produce heat locally in an alternating magnetic field due to magnetic hyperthermia (MH). Since the cancer cells are more sensitive to the local temperature changes than the healthy ones, the MH may result in a larger extent or larger rate of tumor cell death. For this reason, this technique has been proposed to be used as an alternative therapeutic method first in 1957 by Gilchrist et al. [5], but it is still the subject of numerous studies recently.

Nanoparticles intended to be used in biomedicine should fulfil three main criteria: (i) they have to be non-toxic, (ii) biocompatible and (iii) stable in biological milieu. Carbon-based materials, especially carbon nanoparticles are extensively examined for biomedical use, and it has been already found that graphene oxide (GO) is capable of delivering non-water-soluble compounds, e.g., drug molecules to target locations [6]. GO is a hydrophilic, carbonaceous material, with high surface area and tunable pH dependent surface charge properties. Most commonly, graphite oxide is synthesized via oxidation reaction using KMnO_4_ and concentrated H_2_SO_4_, and this synthesis method is industrially scalable [7]. The rich surface chemistry of GO, which allows for an easy application of a wide range of surface functionalization reaction schemes, makes it an excellent platform for the formulation of solid-state nanocomposites or aqueous nanocomposite dispersions [8,9,10,11,12,13,14]. According to the results published by Bai et al. [15], the conductive properties of the GO may have beneficial effect to the hyperthermic efficiency of the magnetic nanoparticles. The GO/MNP composite dispersed in physiological saline solution at 50 mg/mL concentration reached 92.8 °C in 500 s after exposed to 30 kA/m AC (alternating current) magnetic field at 80 kHz. Based on this enormous heat production, the authors suggested the composite for magnetic hyperthermia applications.

While GO/MNP nanocomposites can be obtained under various conditions, almost exclusively in situ methods are used for their preparation, i.e., MNPs are fabricated in the presence of GO lamellae [16,17,18,19,20,21,22,23]. In general, harsh conditions (e.g., strongly alkaline medium, high salt concentration) are applied to produce iron oxide nanoparticles, and the synthesis or work-up is often combined with ultrasonication as well [19,20], which may have a detrimental effect on the carbonaceous sheets. Urbas et al. reported on a composite formation via covalent binding of oleate coated MNPs on activated GO flakes [21]. Multifunctional magnetic nanocarpets delivering doxorubicin (DOX) were synthesized by Sasikala et al. [22], and it was found that the improved antitumor activity combined with enhanced hyperthermic efficiency (23 °C temperature increase in 900 s, at ~12.5 kA/m and ~293 kHz) makes these nanocomposites ideal for cancer theranostics. Electrostatic interaction was used to prepare Fe_3_O_4_@Graphene Oxide composites by Hu et al. [23], building it from GO modified with positively charged polyelectrolyte (poly(diallyldimethylammonium chloride), PDDA) and silica-coated MNPs for environmental purposes, extending thereby the class of magnetically modified supports such as layered silicates [24,25,26,27], mesoporous materials [28,29,30,31] or porous hydrogel matrices [32]. Earlier, we have proposed a mild one-pot synthesis method to fabricate GO/MNP nanocomposites, which was the first of its kind because, unlike other methods based on an in situ particle growth in a GO matrix, it entirely relied on the electrostatic attraction between unmodified GO sheets and bare MNPs [33].

The conductivity of the GO sheets greatly depends on the oxidation degree of the carbonaceous host [34]. The GO particles can be reduced by various methods, either in chemical [35,36], thermal or light-driven ways [37]. Among numerous compounds used for GO reduction, two categories may be distinguished based on the reaction mechanism, in general. The chemicals used for reduction traditionally in synthetic chemistry (e.g., NaBH_4_, LiAlH_4_) form the “well-supported” group, because the mechanism of the reaction is already well-known, while the other compounds applied for GO reduction (e.g., hydrazine, hydroquinone, L-ascorbic acid, etc.) with a not yet fully clarified reaction path belong to “proposed” category [35]. Due to the elimination of the oxygen-containing functional groups in the reduction process, the material changes its color from brown to black, moreover, it becomes more hydrophobic leading to easier aggregation [38]. The efficiency of the reduction can be monitored by the change of the C/O ratio or measuring the conductivity of the carbonaceous compound.

Among various possible methods, Fernández-Merino et al. have carried out the reduction of GO by L-ascorbic acid at 95 °C in various solvents (water, dimethyl formamide, N-methyl pyyrolidone) [39]. They found that the alkaline medium increases the stability of the GO lamellae due to the electrostatic repulsion. Only aqueous reduction was examined by Zhang et al. [40], and it was shown that increasing the concentration of the ascorbic acid the reaction time could be reduced down to few hours starting from 48 h, which was necessary for the completion of the process. A new green synthesis was established by Habte et al. using H_2_SO_4_/H_3_PO_4_ acids and potassium permanganate to prepare and ascorbic acid to reduce graphene oxide with better control on the degree of oxidation [41]. Stepwise reduction was achieved by another research group using Zn powder and sonication to generate hydrogen under acidic conditions [38] by changing the mass of the added solid metal. Reduced GO/MNP composites were prepared mainly for environmental purposes, i.e., for toxic pollution and heavy metal removal using hydrazine as reducing agent [16,42]. In both cases rGO was prepared first, and then the composite was synthesized, adding iron precursors and ammonia solution to initiate the formation of magnetic nanoparticles in situ. According to our knowledge, among the numerous papers published on the synthesis and application of GO/MNP nanocomposite materials, it is only our recent study [33] which demonstrates the theranostic potential of GO/MNP composites prepared by heterocoagulation. However, the effect of post-reduction on the heat evolution caused by these composites in an alternating magnetic field has yet been almost completely disregarded.

Therefore, our main aim is to reveal the effect of reduction conditions on the hyperthermic efficiency of GO/MNP nanocomposites synthesized by the heterocoagulation method. We compare the heat production capabilities of nanocomposites reduced by L-ascorbic acid under acidic and alkaline conditions to those obtained by NaBH_4_ as a commonly applied, strong reducing agent. Heat generation in an alternating magnetic field will be measured for composites with 1/5 and 1/10 GO/MNP mass ratios before and after reduction by NaBH_4_ and LAA at various aqueous suspension concentrations.

## 2. Materials and Methods

### 2.1. Materials

FeCl_2_·4H_2_O, FeCl_3_·6H_2_O, KMnO_4_, NaNO_3_, concentrated H_2_SO_4_, 25 wt% NH_3_ and 30 wt% H_2_O_2_ solution were of analytical grade (Molar Chemicals Ltd., Halásztelek, Hungary) and used without further purification. Reduction was carried out by NaBH_4_ and L-ascorbic acid (Spektrum 3D, Debrecen, Hungary). Constant electrolyte concentration and pH were set and maintained using NaOH, HCl and NaCl in analytical purity (Reanal, Budapest, Hungary). Hydroxylamine, ammonium acetate, glacial acetic acid, FeSO_4_·7H_2_O and 1,10-phenantroline used for spectrophotometric determination of dissolved iron were purchased from Sigma-Aldrich. Ultrapure water produced by a Zeener Power RO&UP system was used as dispersion medium and the experiments were carried out mainly at room temperature (25 °C). Magnetic nanoparticles were synthesized by a traditional co-precipitation method using Fe(II) and Fe(III) salts under strongly alkaline conditions and purified by dialysis against dilute (0.001 M) HCl, as was described in detail elsewhere [4,33]. Graphite oxide was prepared by the Hummers–Offeman method, using KMnO_4_, and NaNO_3_ for oxidation of graphite flakes (SGA20 graphite powder, Kropfmühl GmbH, Germany) under highly acidic circumstances (cc. H_2_SO_4_) [33,43]. The product was purified by dialysis against water to eliminate the excess of salts originating from synthesis.

### 2.2. Synthesis of the Nanocomposites by Heterocoagulation

GO/MNP nanocomposites with 1/5 and 1/10 weight ratios were fabricated by heterocoagulation of oppositely charged GO and MNP at pH ~ 5 [33] as the schematic drawing shows in Figure 1. This simply involved the rapid mixing of the suspensions of GO and MNP in volume ratios which fit to the particle mass concentrations specified in Table 1, keeping them under vigorous stirring for 15–20 min at room temperature. Aqueous samples were then stored undisturbed in a fridge until use.

### 2.3. Reduction of the Composites

GO/MNP nanocomposites with 1/0, 1/5 and 1/10 compositions were reduced using L-ascorbic acid (LAA) and NaBH_4_. First, 100 mL stock suspension of GO/MNP composites was prepared at each GO/MNP ratio with the indicated GO and MNP concentrations (Table 1) corresponding to the 1/0, 1/5 and 1/10 compositions, respectively. Smaller portions of these suspensions were taken and used for further studies; thus, the magnetite concentration was kept constant (5 g/L) in each experiment. The reduction reaction with LAA was carried out by two methods, i.e., under acidic (pH ~ 3.5) and alkaline circumstances (pH ~ 9.3). Room temperature and longer reaction time (4 days) was applied for acidic conditions, and elevated temperature (~95 °C) was used for 30 min in alkaline medium. The concentration of the added organic acid was selected based on previous experiences [44]. The reaction conditions and the sample concentrations are summarized in Table 1.

### 2.4. Experimental Methods

#### 2.4.1. Structure and Morphology

The crystalline structure of synthesized magnetite, graphene oxide nanosheets and GO/MNP nanocomposites were identified by Philips PW 1830 X-ray diffractometer (Philips, Eindhoven, the Netherlands) operating in Bragg-Brentano focusing geometry with CuK_α_ radiation. Powder samples were used for the analysis and the crystallite size of the MNPs was determined from the broadening of the most intensive peak of the XRD pattern by using the Scherrer equation [45]. The morphology of the individual particles and the nanocomposite samples was studied by transmission electron microscopy using a Jeol JEM-1400+ device (JEOL Ltd., Tokyo, Japan) operating at 80 kV accelerating voltage. The primary particle size and the size distribution of MNPs were determined by JMicrovison software version 1.2.7 counting ~100 particles.

#### 2.4.2. Laser Doppler Electrophoresis Measurements

The zeta potential of the magnetite, the GO sheets and their pristine and reduced composite forms were determined in a Nano ZS dynamic light scattering (DLS) apparatus (Malvern Instruments, Malvern, Worcestershire, UK) with a 4 mW He-Ne laser source (λ = 633 nm). Disposable zeta cells (DTS 1070) (Malvern Instruments, Malvern, Worcestershire, UK) were used to record the electrophoretic mobilities at 25 ± 0.1 °C and the Smoluchowski equation was applied to calculate the zeta potentials. The accuracy of the measurements is ±5 mV, and the zeta-standard of Malvern (55 ± 5 mV) was used for calibration. The dispersions were diluted to give an optimal intensity of ~100 cps. The samples were homogenized in an ultrasonic bath for 10 s, after which 2 min relaxation was allowed before the measurements. The electrolyte concentration was kept at constant value (I = 10 mM).

#### 2.4.3. Dynamic Light Scattering Experiments

The average particle size of individual magnetite and GO nanoparticles was determined at 25 ± 0.1 °C using a NanoZS apparatus (Malvern Instruments, Malvern, Worcestershire, UK) operating in backscattering mode at an angle of 173°. The intensity average hydrodynamic diameter (Z-Ave) values were calculated using the second- or third order cumulant fit of the autocorrelation functions, depending on the degree of polydispersity. The individual MNP and GO particles and the various composite samples were studied at the same conditions as in the electrophoresis measurements.

#### 2.4.4. Magnetic Hyperthermia Measurements

The calorimetric measurements were performed using a magneTherm TM (nanoTherics Ltd., Keele, Staffordshire, UK) instrument. The magnetic hyperthermic efficiency was measured at a resonant frequency of 109.4 kHz with magnetic field value of *B* = 24.7 mT by using a 17-turn coil/198 nF capacitor. One milliliter dispersion was tested with a dispersion concentration of 5 g/L for magnetite and 0, 0.5 and 1 g/L for GO. The measurement time was 5 min. The specific absorption rate (*SAR*, W/g magnetite) values for the different field strength values were calculated according to
(1)$$SAR=\frac{C_{p,s}\times \frac{m_{water}}{m_{solid}}}{\frac{\Delta T}{\Delta t}}$$
where $C_{p,s}$ is the specific heat capacity of the medium (water), $m_{water}$ and $m_{solid}$ are the masses of the medium and the nanoparticles, and $\frac{\Delta T}{\Delta t}$ is the rate of temperature change at *t* = 0. For better comparison, the SAR values published in literature are usually related only to the iron content of MNPs instead of the whole mass of nanoparticles.

#### 2.4.5. Determination of Dissolved Iron Concentration

The amount of dissolved iron in the presence of organic acid (LAA) was determined by spectrophotometry applying the protocol recommended by Mykhalyk et al. [46]. Briefly, the composite samples were treated with concentrated HCl first, to dissolve their iron content into Fe^3+^ ions. Then, hydroxylamine and hydrochloric acid are added to the solution in the presence of an ammonium acetate/glacial acetic acid buffer and 1,10-phenantroline was used as indicator. The absorbance of the formed ferroin complex [Fe(o-phen)_3_]^2+^ was measured in acidic media at λ = 510 nm.

## 3. Results

### 3.1. Structural Characterization

The crystalline structure of the synthesized nanocomposites and their starting compounds was studied by X-ray diffraction. A characteristic region of diffraction patterns of the original GO, the magnetite nanoparticles and the 1/5 GO/MNP nanocomposite is presented in Figure 2. At a diffraction angle of 10.4°, graphite oxide shows a sharp diffraction peak revealing its highly ordered layered structure composed of parallelly oriented graphene oxide particles, which are well observable in the corresponding TEM image in Figure 3. The position of this (002) reflection usually varies strongly with the ambient humidity conditions and spans the c-axis repeat distance range of ca. 6–9 Å. Another band also appears in the X-ray pattern of graphite oxide: the second one usually has definitely lower intensity, but its 2Θ angle is twice as large as for the first one. However, since the latter is a second order “overtone” band (originating from interference of wave trains reflecting from the lattice planes with a distance shift of two wavelengths), both bands refer to the same periodical distance of ~8.5 Å. Magnetite usually represents six characteristic peaks corresponding to the spinel structure of the iron oxide sample. The reflections at 30.34° and at 35.4° are the most intense bands belonging to the (220) and (311) lattice planes of face-centered cubic structure. The diffractogram of the 1/5 GO/MNP composite resembles more the magnetite than the GO sample showing the same characteristic reflections at 30.34° and at 35.6°. Since the magnetite phase is dominant in the composite samples, magnetite-related features are highly expected in their XRD pattern as well. However, the total absence of the graphite oxide-related band is a very remarkable feature of the pattern of GO/MNP, because one could justifiably expect a contribution from such a strong and narrow GO reflection in a sample that contains more than 15 wt% of GO. One may assume that the absence of this band is due to the leaching of the carbonaceous sheets upon synthesis, which results in a significant deviation from the original, 1/5 mass ratio. However, we need to point out that, unlike for many other synthesis protocols, the mass ratio of the two phases cannot change upon the heteroaggregation process, because there is virtually no loss of particles (e.g., via leaching, chemical transformation), and thus the mass of the nanocomposite equals to the mass of NMP and GO mixed together in a predetermined amount. Therefore, the lack of the ~8.5 Å basal spacing in the nanocomposite samples clearly substantiates that the carbonaceous sheets have become highly dispersed (exfoliated) upon the synthesis process. This strong assumption is justified if one considers that a randomly distributed and oriented platelet-structure composed of individual graphene oxide particles can be retained in the coherent composite structure even after the removal of the solvent (water) content of the system. As a result, while the amount of GO in the composite powder is still relatively high (ca. 15 wt%), it is not present as stacked, multi-layered particles that could give rise to a strong X-ray reflection. The average crystallite size of the magnetite counterpart calculated by the Scherrer equation for the most intense reflection at 35.6° 2Θ is ~9 nm, which is somewhat smaller than the primary particle diameter determined by the TEM images (~10 nm). This latter indicates a common phenomenon for SPIONs (superparamagnetic iron oxide nanoparticles): a thin amorphous layer is present on the MNP’s surface, which does not show the crystallite structure characteristic to the core of the particle.

Transmission electron microscopy images of some selected samples (GO, MNP, 1/5 GO/MNP composite before and after reduction) are presented in Figure 3 for comparison. The GO sample consists of folded paper-like layers, indicating the highly delaminated, single-layer state of graphene oxide. The average primary particle size of individual magnetite nanoparticles (top panel, right) was 9.5 ± 1.2 nm, which remained almost unchanged in the pristine GO/MNP composite (9.4 ± 1.3 nm, bottom panel left), while it became larger with slightly broader size distribution due to the reductive treatment (12.6 ± 2.3 nm, bottom panel right). As can be seen, the reduction process changes the morphological attributes of GO/MNP composites, because the distribution of the MNPs greatly changed. Before the reduction reaction, the magnetic nanoparticles are more or less well-distributed showing an average size of ~9.5 nm (bottom panel, left), but afterwards, the nanomagnets form bigger clusters according to the image of the rGO/MNP taken at lower magnification in Figure 3 (bottom, right). This change can be interpreted by considering the modification of the surface charging of GO during the reduction procedure. Lowering the oxidation degree of GO leads to diminution of the functional groups resulting in lower surface charge on the carbonaceous lamellae at the same pH [34,47]. Since the interaction between the magnetite nanoparticles and GO sheets is based on electrostatic attraction [33], changes of the composite’s character might be expected after reduction. Here, we did not observe lowering of the iron oxide content in the rGO/MNP composite, as can be seen in Figure 3, only the modified distribution of the MNPs was detected.

### 3.2. Surface Charge Characteristics

The pH-dependent surface charging of GO and magnetite nanoparticles was investigated by electrophoretic mobility measurements in order to find the optimal pH range for the composite formation by heterocoagulation process. The calculated zeta potential (ZP) values of the nanomagnets and GO particles are presented in Figure 4. Graphene oxide sheets exhibit negative charge, the ZP is lower than −30 mV, over the whole pH range studied due to the dissociation of the acidic functional groups. The highly negatively charged surface usually leads to long-term kinetic stability, but in the case of GO the relatively large extension of lamellae in two dimensions (equivalent hydrodynamic spherical diameter ~800 nm) results in particle sedimentation. Magnetite nanoparticles are amphoteric type of materials, which means that the surface charge is changing depending on the pH of the medium. MNPs are highly positively charged (~40 mV, Figure 4, circles) at low pH, while they become negatively charged in the alkaline pH range after a charge reversal at pH ~ 8, where the MNP surface is neutral [48]. Varying amounts of negative charges develop on GO lamellae depending on the dissociation of acidic functional groups in the studied pH range [43,47]. The measured zeta potential descends to −50 mV, when the GO reaches its fully deprotonated state (Figure 4, open squares). In order to maximize the electrostatic attraction between GO and MNP but keeping the pH close to the physiological one, the composite formation process was carried out at pH ~ 5.5 where the compounds are highly and oppositely charged, as indicated by the red-highlighted region in Figure 4.

Regarding the possibility to monitor the progress of the reduction process, we considered using two main methods to provide data that would allow comparison of each stages of reduction for the individual carbon phases and their composites. In the case of pure graphene oxide, the C/O ratio has long been used to follow and compare the efficiency of the reduction [49]. Unfortunately, for the present system, the C/O ratio will not provide the sought data, because the reduced composite samples still contain magnetite and some organic compound as well. Since these two materials contain oxygen and carbon too, which may also transform during the reduction step influencing the amount of carbon and oxygen, the C/O ratio would not be informative. Conductivity measurements are also used to monitor the oxidation degree of graphene/graphene oxide samples [50], but grain boundary issues and the presence of organic compounds may also have disturbing effect, as in the case of C/O ratio.

### 3.3. Heat Generation in Nanocomposite Dispersions

The heat production of the pristine and the reduced GO/MNP nanocomposites with various compositions was studied by calorimetric measurements. Time-resolved temperature measurements were performed, and the original magnetite (MNP) and the reduced graphene oxide (rGO) dispersions were used as reference. In our previous work [33], we demonstrated that the 1/50 GO/MNP composite aqueous dispersions exhibited even higher heat production than the pure magnetite sample, likely due to the contribution of Joule loss generated by the GO sheets. In that case the amount of GO was almost negligible (less than 2 wt%) compared to the mass of MNPs. Since we intended to investigate the composition dependency of hyperthermic efficiency at much larger GO contents, we have chosen the 1/5 and 1/10 GO/MNP ratios in the present study.

#### 3.3.1. GO/MNP Composites before Reduction

First, we determined the heat production of the pristine composites at 1/5 and 1/10 GO/MNP compositions, and, as the left panel of Figure 5 shows, no difference between the samples could be observed. The composites exhibited almost the same rate of temperature increase, the calculated SAR was 9.2 and 8.8 W/g, which was significantly higher than that of the GO sample and it almost reached the level produced by the magnetite itself (11.6 W/g). Since all dispersions contained the magnetic nanoparticles at equal concentrations (5 g/L), the deviation should be assigned to the presence of the GO lamellae. The minor deviation (only 0.5 °C) in the final Δ*T* between pure magnetite and the composite samples indicates that the GO sheets do not attenuate the heat production on a significant scale. As the pure GO sample also produces some moderate temperature rise in 300 s (less than 3 °C), two possible processes can be considered. One may originate from the electrolyte content of the aqueous dispersion of the GO and the second one is likely due to the resistance heating effect caused by Joule-losses of graphene oxide particles as we suggested in our previous work [33]. A GO/MNP nanocomposite of nearly 1/1 mass ratio was proposed by Bai et al. as thermoseeds in magnetic hyperthermia treatment of cancers, based on their excellent heating abilities [15]. This magnetic nanomaterial dispersed in physiological saline solution reached 92.8 °C in 500 s. Such a colossal temperature rise is indeed quite promising for biomedical application but considering the very high suspension concentration (50 mg/mL) and the strong magnetic field applied in that publication (30 kA/m), the heat generation of the composites prepared in our study is comparably efficient. Although the SAR value (~24 W/g) estimated for that case is still higher, compared to our measurements recorded at 19.5 kA/m (24.7 mT) in dispersions containing 5 mg/mL MNP, our composites also show promising hyperthermic potential by taking into account the quadratic dependence of the heat production from the magnetic field strength [4]. Besides the high particle concentration, which is ten times larger than ours, the electrolyte content of the medium (physiological saline solution was used in this case) probably also contributes to the measured enormous temperature rise.

#### 3.3.2. Heat Production of rGO/MNP Composites Obtained by Borohydride Treatment

On the right side of Figure 5, the temperature differences are presented for the MNP, GO and GO/MNP composites measured after the reduction using NaBH_4_, which is a frequently used compound for GO reduction [35,37]. The picture is definitely different compared to the pristine samples (Figure 5, left side). After the reduction, both the 1/5 and 1/10 composites showed significant heat production: the respective SAR values are 12.7 and 16.2 W/g (Table 2) calculated by considering only the initial part of the heating curves, where the slope is slightly lower for the 1/5 GO/MNP. If we consider the whole interval, this variance almost disappears, and hardly different SAR values (20.0 and 18.8 W/g) can be obtained. The heat production of MNP does not change significantly as a result of the reduction: the final Δ*T* is less only by 0.2 °C after treating the sample with NaBH_4_. We note that the reduced composites were not stable: the solid material started to sediment almost immediately after addition of the reducing agent. Although the borohydride reduction of GO/MNP composites apparently improved their magnetic hyperthermia applicability, it also raises stability issues. This problem poses the argument for favoring milder reducing agents, such as the frequently used ascorbic acid [35,41].

It is known that the magnetic properties of nanoparticles play an important role in their behavior in an alternating magnetic field [51]. An inverse relationship between the SAR value and the average particle size was published earlier for MNPs [52], but the colloidal stability was not considered carefully. Later, this finding was supplemented by taking into account the aggregation properties of the magnetic nanoparticles, i.e., counting on the dipole–dipole and the exchange interactions as well [51]. Since both the dipole–dipole interaction and the exchange coupling is influenced by the interparticle distance, it is not surprising that any slight change in the colloidal state, which is related to the particle interactions, may lead to the notable alteration of magnetic properties. If the particles are able to approach each other closer, a collective magnetic state might be created, where the moments will be randomly oriented or aligned to the AC field. This cooperative effect among magnetic nanoparticles may lead to more pronounced hyperthermic effect, but if the particle aggregates become too large, they start to sediment, thereby disappearing from the observed area in which the measurement is carried out. However, for these samples, the stability issue is even more complicated, since the magnetic nanoparticles interact with the carbonaceous lamellae; furthermore, any organic compounds present in the dispersion medium may have an influence on the colloidal stability, i.e., the spatial distribution of the particles as well. In our case the heating curves are not perfectly straight lines, but they do show linearity within broad intervals of magnetic field strength. Despite the careful isolation of the sample from the environment, we cannot prevent some heat loss, which is normal for non-adiabatic calorimetric measurements and makes it more realistic [53]. The slight bending on several heat curves may be due to the change in particle distribution inside the measuring vial. If the GO loses some functional groups during the reduction process, the decreased surface charge of lamellae may result in a change of the planar distribution of MNPs, leading to their clustering similar to that observed on the TEM image of the LAA-reduced 1/5 rGO/MNP sample (Figure 3).

#### 3.3.3. Heat Production of rGO/MNP Composites Obtained by Ascorbic Acid Treatment

According to the method applied earlier [39], we carried out the reduction in alkaline media, by adding small portion of ammonia to the mixture of GO/MNP and LAA. In parallel, the reduction process was performed also in acidic conditions. The final pHs of the mixtures were 9.3 and 3.5, respectively. Figure 6 shows how the different reaction conditions affect the hyperthermic efficiency of the studies materials and the calculated SAR values are summarized in Table 2.

Under acidic conditions, the heat production of rGO and 1/5 rGO/MNP samples becomes worse than that of pristine MNP and only the sample of 1/10 composition shows the same efficiency. The calculated SAR value was 11.8 W/g for the 1/5 rGO/MNP, while the original MNP exhibited 11.6 W/g. The difference between 1/5 and 1/10 samples may be attributed to the low pH, where the LAA is partially protonated, and its amount is likely not enough to reach the same degree of reduction for the 1/5 sample with double GO content. Contrary to this, the alkaline conditions significantly improved the temperature rise measured for the composites, especially in the case of 1/5 rGO/MNP, where the SAR value increased from 9.2 up to 20.3 W/g after reducing it by LAA/NH_3_ mixture.

The heat dissipation efficiency in the AC magnetic field strongly reflects not only the propensity of magnetic dipoles, but also their different types interactions. Although SAR values decrease with aggregation at first glance, Ovejero et al. [54] have revealed that inter-aggregate dipolar interactions cause an enhancement in the magnetization, while intra-aggregate dipolar interactions lead to a decline (demagnetization). Therefore, any change in the spatial distribution of magnetic dipoles (i.e., the nanomagnets) may result in different dynamic magnetic properties (e.g., hysteresis loop). In the case of GO/MNP nanocomposites, the surface densities of MNPs on GO lamellae can be tuned by their mass ratio. As we have demonstrated in an earlier publication, the 1/50 GO/MNP nanocomposite exhibited even higher heat production than the pure magnetite sample [33], which may be the consequence of the inter-aggregate dipolar interactions between surface-bound MNPs. The performance of 1/10 and 1/5 GO/MNP composites was far from that of 1/50 sample. We tried to improve them by reducing GO, which may lead to the clustering of MNPs on rGO surface as seen on the TEM image of the 1/5 rGO/MNP sample (Figure 3).

The 1/10 rGO/MNP sample also presented slightly higher heat production, but the SAR increase was only 2.3 W/g compared to the pristine 1/10 GO/MNP sample, reaching 11.6 W/g for the reduced composite. It is interesting to note, that the amount of added LAA was equal in all cases, but the mass of the GO changed, since the MNP concentration was kept constantly at 5 g/L. In the 1/10 composite, the amount of the carbonaceous host was half compared to the 1/5 sample, so the LAA could reduce the GO to a greater extent, leading to almost complete transformation from GO to graphitic carbon. As in the case of NaBH_4_, due to the overly strong reduction and the large excess of reductant, the treatment resulted in weaker heat producing samples, which is contrary to the aim of this study. Thus, we investigated the effect of the added amount of LAA on the hyperthermic efficiency of the reduced composites next.

The heating curves recorded for the 1/10 rGO/MNP composites treated by LAA in various concentrations at pH 9.3 are presented in Figure 7. At first glance, the samples behave in a slightly different way in AC field following the reduction with 5 and 7.5 mM LAA than that reduced with 2.5 mM LAA. The hyperthermic performance of the 1/5 rGO/MNP composites were not influenced by the added LAA amount used in the reduction stage of its synthesis, while in the case of the 1/10 ratio the lowest reductant concentration (2.5 mM) leads to much lower heat production. The latter exhibits a temperature increase (ca. 3.8 °C) which is smaller even than that found for the pristine 1/10 GO/MNP sample (Figure 5, left). Based on this observation, a concentration threshold of ca. 5 mM can be established regarding the amount of LAA necessary to maintain a fairly good hyperthermia performance for the 1/10 composition. However, while both MNP/GO samples develop a practically relevant heat production capability upon borohydride reduction, it seems that only the nanocomposite containing larger amounts of GO (1/5 GO/MNP mass ratio) bears a noticeable increase in the heating rates as compared to the naked MNP particles. This finding, however, does not justify the greater practical utility of the nanocomposite dispersion obtained by borohydride assisted reduction, because the biomedical applicability may also be influenced by the size features of the dispersed heat-generating particles and their distribution in the aqueous medium. We have demonstrated in an earlier study that, depending on the GO/MNP mass ratio, the hydrodynamic diameter of the composites may change drastically [33]. Herein, due to the extensive aggregation found for the samples reduced by NaBH_4_, DLS was inadequate for providing a reliable estimate of the average size of aggregates in that system. Therefore, a meaningful comparison can only be provided by visual observation of the composite dispersions. These substantiate that the LAA-reduced samples remain dispersed in water for a much longer time as compared to borohydride-reduced composites, which form large flocculated particles reminiscent of light carbon soot materials obtained, e.g., upon pyrolysis of graphite oxide. It is clear then that the added LAA behaves not only as a reducing agent, but it contributes to the particle stabilization as well.

Based on our previous experience [44] and due to the applied strong acidic and alkaline conditions we tested the probable dissolution of the ferrous ions from the crystallite structure of the magnetic core. The iron content was measured by spectrophotometric method [46], and the amount of the dissolved iron was found to be below the detection limit, which is ca. 0.5 mg/mL in our case. It can be concluded that neither the acidic nor the alkaline reduction promoted the dissolution of the solid matrix during the treatment of GO/MNP composites with LAA.

## 4. Discussion

### 4.1. Perspectives on Magnetic Hyperthermia of rGO/MNP Composites

Relatively few papers [55,56,57] were published on the possible application of graphene oxide/magnetite nanocomposites for magnetic hyperthermia, and only one was found related to the study of the heat production of reduced GO/MNP in an AC magnetic field [58]. Rodrigues et al. reported on interesting multifunctional graphene-based magnetic nanocarriers for combined hyperthermia and drug delivery [55]. They found high heating efficiency (SAR, >300 W/g) and dual pH and thermal stimuli-responsive drug-controlled release under an alternating magnetic field (*f* = 340 kHz, *H* = 21.0 kA/m) of the graphene-based yolk-shell magnetic nanoparticles functionalized with copolymer Pluronic F-127 and loaded with doxorubicin drug molecules. Although this nanostructure is really promising for cancer treatment, the studied composite material was prepared by a complex multistep synthesis method unlike our composite material, which involves no functionalizing agents or loaded molecules. GO/MNP nanocomposites of various compositions were synthesized in situ for in vitro apatite mineralization by Miyazaki et al. [56]. In harmony with their previous results, both the osteoconductivity and the magnetically induced thermal response studied in agar phantom improved at higher Fe_3_O_4_ content of the nanocomposite. Albeit reduced GO/MNP was discussed throughout the paper, none of the broad classes of reducing agents were directly applied, only the possible decomposition (resulting in a loss of oxygen content, which is often misinterpreted as a true chemical reduction) of GO during the co-precipitation of magnetic nanoparticles was assumed. A solvent evaporation method was applied by Sugumaran et al. to reside iron oxide nanoparticles of various core sizes in the GO sheet based host [57]. The PEGylated GO/MNP composites with the largest MNP core size showed not only outstanding hyperthermic efficiency with SAR value higher than 5000 W/g, but they exhibited excellent colloid stability as well. Finally, the sole previous study focusing on naked rGO/MNP samples was published by Gupta et al. [58] using hydrazine to chemically reduce the composite following its synthesis process via co-precipitation. However, their study differs from ours in many respects such as (i) the type of reducing agent, (ii) the synthesis pathway (classical in situ co-precipitation vs. heterocoagulation) and, most importantly, (iii) their composite material was also loaded with drug molecules to enhance the magnetic thermotherapy. Despite their finding that the DOX-loaded rGO/MNP showed significant enhancement in the synergistic antitumor therapeutic efficacy in the AC magnetic field, the heat production slightly decreased related to the pristine magnetite.

### 4.2. Utility of rGO/MNP Composites Synthesized in This Study

Compared to the aforementioned studies, the present is unique in terms of the fabrication of the bare rGO/MNP composites directly employed for magnetic hyperthermia. Contrary to the exclusively used in situ co-precipitation, we successfully applied heterocoagulation to prepare GO/MNP nanocomposites from separately synthesized and purified GO and magnetite dispersions. The mild method allows us to use only electrostatic attraction to reside the MNPs on the carbonaceous host without applying harsh chemical circumstances. Aiming to achieve only partial reduction of GO content in the composites, the strong reducing agent, the NaBH_4_, was found to be too aggressive of a compound for this purpose, although the heat production apparently slightly increased but the colloidal stability of the reduced samples diminished substantially. Depending on the reaction conditions (e.g., pH, reactant concentration), LAA revealed excellent ability for partial reduction of GO sheets into rGO form leading to increased thermal response in moderate AC magnetic field (109.5 kHz, 24.5 mT). For fine tuning of the hyperthermic efficiency by changing the oxidation state of the carbonaceous host, the LAA proved to be a good candidate, due not only to the enhanced thermal response of 1/5 rGO/MNP obtained by alkaline reduction, but also to the improvement of the colloidal stability. Since this latter property, among other criteria (biocompatibility, non-toxicity, etc.), is crucial for biomedical application but aggregation properties have yet barely been studied in relation to magnetic hyperthermia of MNPs and their composite materials, we believe that the further optimization of this method to afford high-stability efficient nanocomposites has high potential for cancer theranostics.

## Figures and Tables

**Figure 1 nanomaterials-10-02426-f001:**
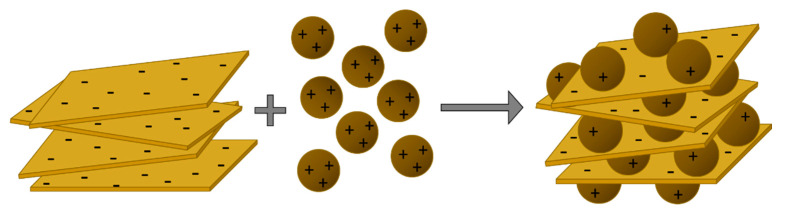
A schematic representation of the formation of graphene oxide/magnetite (GO/MNP) nanocomposites by heterocoagulation of the oppositely charged nanoscale counterparts.

**Figure 2 nanomaterials-10-02426-f002:**
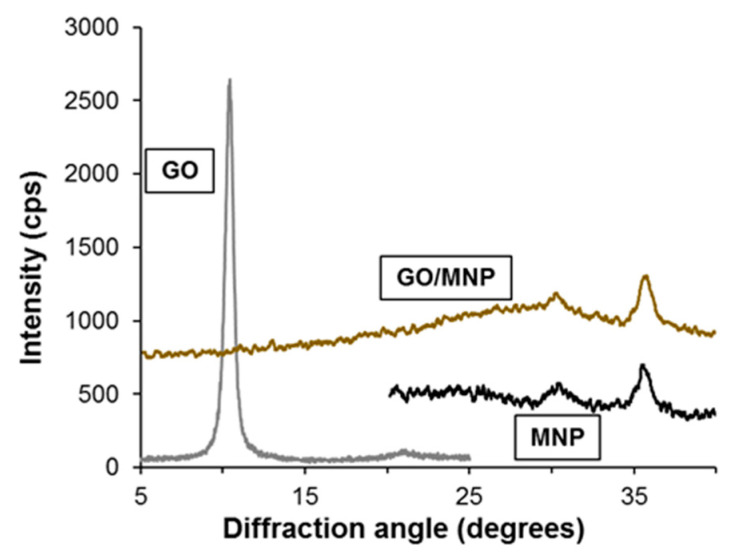
XRD patterns of individual MNP and GO nanoparticles and of the 1/5 GO/MNP nanocomposites in powder form.

**Figure 3 nanomaterials-10-02426-f003:**
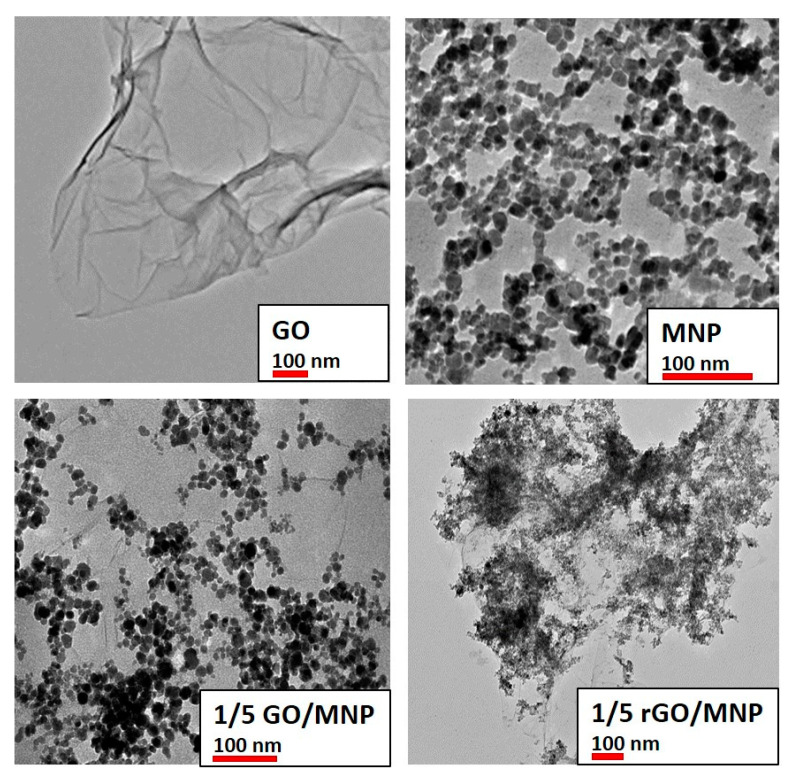
TEM images of the individual nanoparticles and the 1/5 GO/MNP nanocomposites before and after reduction with LAA. The red scale bar corresponds to 100 nm to assist better visibility and easier comparison.

**Figure 4 nanomaterials-10-02426-f004:**
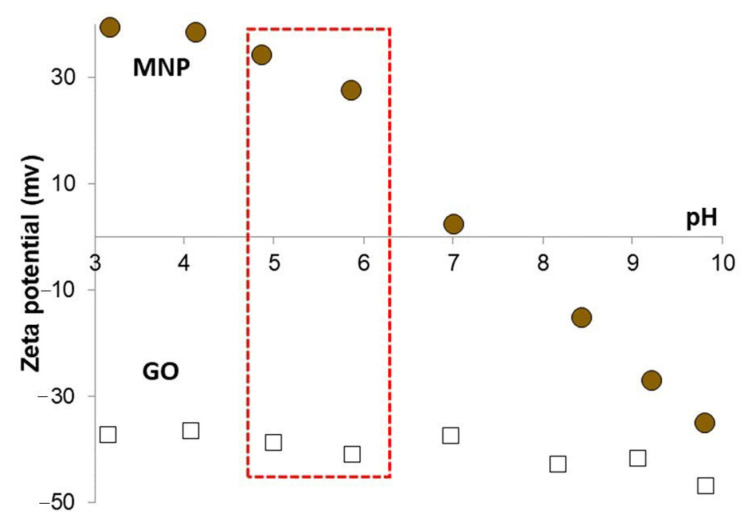
Variable surface charging of MNP and GO as the function of pH in the presence of 10 mM NaCl used as background electrolyte.

**Figure 5 nanomaterials-10-02426-f005:**
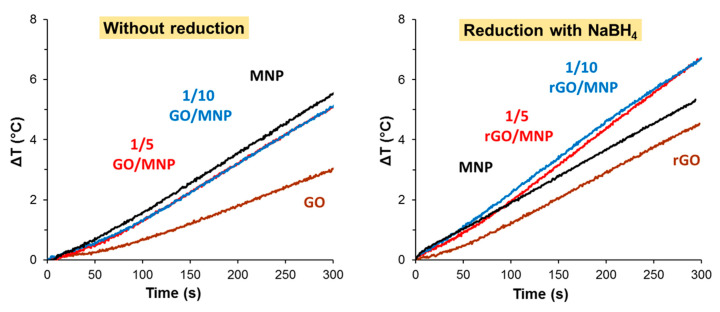
Heat production of pristine GO/MNP nanocomposites (**left**) and after using NaBH_4_ as reducing agent (**right**) measured in the AC field at 109.4 kHz and at 24.5 mT.

**Figure 6 nanomaterials-10-02426-f006:**
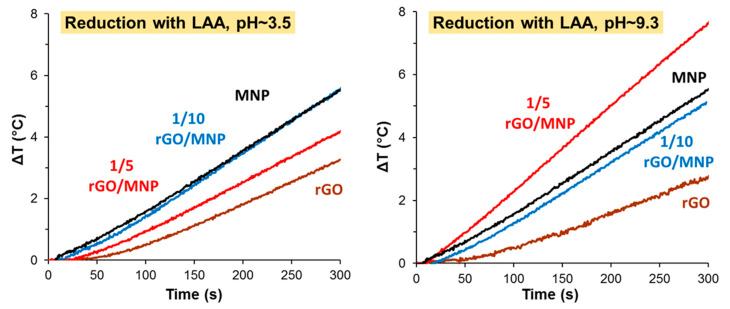
Heat production of GO/MNP nanocomposites reduced by 5 mM LAA under acidic (pH ~ 3.5) and under alkaline conditions (pH ~ 9.3).

**Figure 7 nanomaterials-10-02426-f007:**
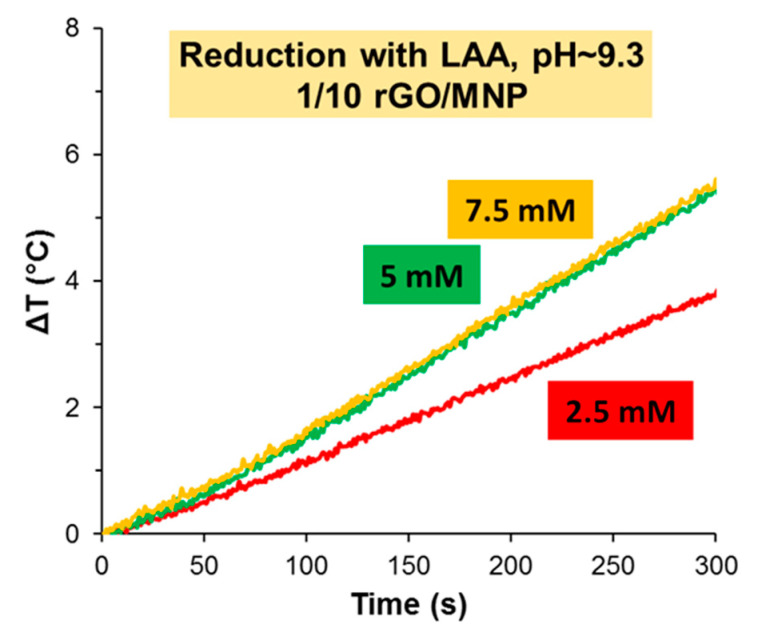
Heat production of 1/10 rGO/MNP nanocomposites applying LAA at various concentrations (2.5, 5 and 7.5 mmol/L) at pH 9.3.

**Table 1 nanomaterials-10-02426-t001:** Reduction reaction conditions and sample concentrations.

	NaBH_4_	Acidic (pH ~ 3.4–3.7) ~25 °C, 4 days	Alkaline (pH ~ 9.2–9.5) ~95 °C, 30 min
**GO/MNP 1/0** GO:1 g/L, MNP: 0 g/L	50 mM	LAA 5 mM	LAA: 5 mM, NH_3_: 5 µL/mL
**GO/MNP 1/5** GO: 1 g/L, MNP: 5 g/L	50 mM	LAA 5 mM	LAA: 5 mM, NH_3_: 5 µL/mL
**GO/MNP 1/10** GO: 0.5 g/L, MNP: 5 g/L	50 mM	LAA 5 mM	LAA: 5 mM, NH_3_: 5 µL/mL

**Table 2 nanomaterials-10-02426-t002:** Summary of calculated specific absorption rate (SAR) values (in W/g) obtained for GO and GO/MNP samples.

	Without Reduction	After Reduction with		
		NaBH_4_	LAA, pH ~ 3.5	LAA, pH ~ 9.3
GO	3.6	7.1	2.6	1.9
1/5 GO/MNP	9.2	12.7	7.6	20.3
1/10 GO/MNP	8.8	16.2	11.8	11.6

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
