# Peer review of "Tunable Magnetic Hyperthermia Properties of Pristine and Mildly Reduced Graphene Oxide/Magnetite Nanocomposite Dispersions"

_nanomaterials, 2020, doi:10.3390/nano10122426_

Round 1

Reviewer 1 Report

In the manuscript “Tuneable magnetic hyperthermia properties of pristine and mildly reduced graphene oxide/magnetite nanocomposite dispersions” Illés et al. investigate the theranostic potential of GO/MNP composites prepared by heterocoagulation, with particular attention at the effect of post-reduction on the heat evolution when these composites are subjected to an alternating magnetic field.

Even though the materials and the preparation method are not entirely new, the paper is well written, all the experimental procedures are carefully described, and the analysis and discussion of the results is well clarified. It can be of interest to the specialized reader.

I can therefore recommend publication after these points are addressed:

  • The authors state that the reduction process changes the morphological attributes of GO/MNP composites, because the distribution of the MNPs greatly changed, while the carbonaceous host remained intact. However it seems that the size of the MNP is reduced too. The authors should possibly show the TEM images at the same magnification, so to allow a better comparison among them.
  • On a similar point, it seems that the size distribution of MNPs alone and MNPs in the composite are also different (fig 3 top right and bottom left). Do they have an explanation for that? Is the formation of the nanocomposite also modifying the nanoparticles themselves?

Author Response

Responses to both reviewers' comments are included into the same coverletter.

Reviewer 2 Report

The paper is devoted for RGO/magnetic nanoparticles composites preparation and their investigations for hypertermia applications. The topic is generally interesting, however the paper contains unexplained places (below) and need major revisions.

What is the impact of rGO on magnetic hypertermia performance?

Why is needed the chemical modification of your composites by NaBH4 and LLA? Which physical parameter (responsible for hypertermia performance) is changing by these chemical modifications and in which way?

How is possible to control hypertermia performance of your composites by their chemical modifications?

What can be the impact of mass ratio RGO/magnetic nanoparticles for composite hypertermia performance?

Author Response

(The authors gave the same response as above.)

Round 2

Reviewer 2 Report

Authors make proper corrections and I suggest publish the paper as is.